# CAR-T Cell Therapy in Ovarian Cancer: Where Are We Now?

**DOI:** 10.3390/diagnostics14080819

**Published:** 2024-04-16

**Authors:** Clare Cutri-French, Dimitrios Nasioudis, Erin George, Janos L. Tanyi

**Affiliations:** 1Department of Obstetrics and Gynecology, University of Pennsylvania Health System, Philadelphia, PA 19104, USA; clare.cutri-french@pennmedicine.upenn.edu; 2Division of Gynecologic Oncology, University of Pennsylvania Health System, Philadelphia, PA 19104, USA; 3Moffitt Cancer Center, Richard M. Schulze Family Foundation Outpatient Center at McKinley Campus, 10920 McKinley Dr, Tampa, FL 33612, USA

**Keywords:** ovarian cancer, CAR-T therapy, target antigens, mesothelin

## Abstract

The success of chimeric antigen receptor T-cell (CAR-T) therapies in the treatment of hematologic malignancies has led to the investigation of their potential in the treatment of solid tumors, including ovarian cancer. While the immunosuppressive microenvironment of ovarian cancer has been a barrier in their implementation, several early phase clinical trials are currently evaluating CAR-T cell therapies targeting mesothelin, folate receptor a, HER2, MUC16, and B7H3. Ongoing challenges include cytokine-associated and “on-target, off-tumor” toxicities, while most common adverse events include cytokine release syndrome, hemophagocytic lymphohistiocytosis/macrophage activation-like syndrome (HLH/MAS), and neurotoxicity. In the present review, we summarize the current status of CAR-T therapy in ovarian cancer and discuss future directions.

## 1. Introduction

Despite advancements in chemotherapy and surgery, ovarian cancer remains one of the most lethal gynecologic malignancies. Due to the late onset of symptoms and the lack of screening modalities, ovarian cancer is often diagnosed at advanced stages, when curative treatment options are limited. The majority of patients experience a recurrence and ultimately succumb to their disease. In recent years, cell-based immunotherapies have emerged as promising options for these patients with otherwise limited treatment options [1]. Specifically, clinical trials have investigated checkpoint inhibitors, including pembrolizumab, cancer vaccines, and adoptive cell therapy using tumor-infiltrating lymphocytes, with overall limited efficacy and disappointing results [2,3]. Among immunotherapies, chimeric antigen receptor T-cell (CAR-T) therapy has demonstrated exceptional results in hematologic malignancies with six FDA-approved therapeutics and also shows promise as a novel therapeutic option in ovarian cancer [4,5].

## 2. CAR-T Design

In CAR-T cell therapy, a patient’s T cells are genetically modified to express a synthetic receptor that specifically recognizes and directs T cells to cancer surface antigens, enhancing precision and efficacy [6]. This leads to an immune response that is independent of the major histocompatibility complex (HLA)-antigen, avoiding certain tumor escape mechanisms such as MHC-1 downregulation [7,8]. Once activated, T cells release cytokines such as interferon- γ, perforin, and granzyme to induce cell lysis.

CAR-T cells contain three main components: an extracellular antigen recognition domain, a transmembrane domain anchoring the receptor in the T cell membrane, and an intracellular domain responsible for triggering T cell activation (Figure 1) [6]. The extracellular recognition domain displays single-chain variable fragments (scFv) formed from the heavy and light chains of an antibody fused through a flexible linker [9]. A spacer (or hinge region) connects the antigen recognition domain to the transmembrane domain while providing length and flexibility. It typically comprised fragments from IgG1, IgG4, or CD8α [10]. The intracellular domain contains signal pathway molecules responsible for activating T cells, including the CD3-ζ chain or the FcγRI protein.

Continuous advancements in CAR-T cell engineering aim to optimize these components, improve tumor targeting, minimize off-target effects, and enhance therapeutic efficacy. The first-generation CAR-T cells utilize only a CD3-ζ signaling domain, without a second co-stimulation domain, and do not provide sustained responses in vivo or in clinical work. Second and third-generation CAR-T cells include co-stimulatory domains to allow for T cell activation. This co-stimulation dramatically increases the persistence of engineered T cells and improves therapeutic duration [11,12]. Fourth-generation CAR-T cells incorporate additional features such as the ability to produce cytokines (IL-2 and IL-12) or antibodies [13].

Several challenges exist in the use of CAR-T cell therapy in ovarian cancer. In comparison to the hematologic malignancies, there is significant intra- and inter-tumor heterogeneity, making it difficult for a single CAR-T cell to effectively target all tumor cells. There is the potential for tumor cell survival through the loss or downregulation of antigen expression in a phenomenon known as antigen escape [14]. Careful consideration is required to minimize on-target, off-tumor CAR-T cell-mediated toxicity due to target antigen expression on non-cancerous cells [15].

Additionally, the immunosuppressive nature of the tumor microenvironment makes the use of CAR-T cells more difficult in ovarian cancer [16]. The tumor microenvironment includes the vasculature, extracellular matrix, stromal, and immune cells. Specifically, immunosuppressive cells include tumor-associated macrophages, myeloid-derived suppressor cells, and regulatory T cells. These cells release anti-inflammatory cytokines (i.e., IL-6, IL-10, TFGβ) and induce the production of reactive oxygen species, which inhibits the activity of CAR-T cells [14,17]. T cell inhibitory surface receptors like PD-1 and LAG3 are upregulated [18]. Tumor stroma can also limit the ability of CAR-T cells to infiltrate solid tumors [19].

## 3. Target Antigens

Ovarian cancer, characterized by its diverse histological subtypes and a high degree of molecular heterogeneity, presents a complex landscape of potential target antigens for CAR-T cell therapy. Identifying and selecting suitable antigens is crucial for the success of CAR-T therapy. Ideal targets are tumor-associated antigens that are highly expressed on the surface of cancerous cells while having minimal expression on other tissues to minimize on-target, off-tumor effects. Several prominent tumor-associated antigens have been explored in the context of ovarian cancer, each with unique considerations and challenges. A summary of target antigens and expression patterns in normal tissues is presented in Table 1.

### 3.1. Mesothelin

Mesothelin is a cell surface glycoprotein expressed in pancreatic, ovarian, mesothelioma cells, and squamous epithelium [20]. It has been shown to bind to CA-125 [21]. Though its biologic function is not clearly defined, it may play a role in cell adhesion and tumor metastasis [22]. The high expression of mesothelin in ovarian cancer, particularly serous subtypes, makes it an attractive target for immunotherapies [23]. Several preclinical studies have demonstrated mesothelin’s safety and feasibility as a target for CAR-T cells [20].

### 3.2. Folate Receptor Alpha

Folate is essential for cell division. Folate receptor alpha (FRα) is a glycosylphosphatidylinositol (GPI)-anchored membrane protein that is highly expressed in epithelial ovarian, breast, and lung cancers [24]. There is limited expression in normal tissue, including the lungs, kidneys, choroid plexus, intestines, heart, and placental tissue [25]. In normal tissues, FRα is typically restricted to the apical surfaces of organs, away from the circulation [26]. FRα is overexpressed in 80% of epithelial ovarian cancers. It is considered a marker of tumor aggressiveness, as it correlates with histologic grade and stage [27,28].

### 3.3. Mucin 16

Mucin 16 (MUC16) is a mucin glycoprotein frequently overexpressed in epithelial ovarian cancer. Through its interactions with mesothelin, it contributes to tumor metastasis [29]. Its extracellular structure includes a cleavage site, with a cleavable domain (CA-125) and a retained domain (MUC16ecto). CA-125 is a clinically established biomarker for ovarian cancer. The presence of soluble CA-125 in the bloodstream poses additional challenges to the effective targeting of tumor cells. The retained extracellular MUC16ecto domain of the antigen is an attractive target for CAR-T cell therapy. It is also expressed in normal tissues of the uterus, endometrium, fallopian tubes, and ovaries [30].

### 3.4. Human Epidermal Growth Factor Receptor 2

Human epidermal growth factor receptor 2 (HER2), also known as ERBB2, is a transmembrane tyrosine kinase receptor overexpressed in several solid cancers including ovarian cancer [31]. It is the target of the monoclonal antibody trastuzumab. Several CAR-T designs targeting HER2 have been developed, and trials are ongoing in other solid tumors. One clinical trial is recruiting ovarian cancer patients (NCT04511871).

### 3.5. Other

Additional tumor antigens have been investigated in vitro and in vivo but do not yet have clinical data [3]. These include claudin 6 [32,33], protein tyrosine kinase 7 (PTK7) [34,35], epithelial cell adhesion molecule (EPCAM) [36], Annexin A2 (ANXA2) [37], anti-Mullerian hormone receptor type 2 (AMHR2) [38,39], tumor-associated glycoprotein 72 (TAG72) [40], programmed cell death ligand 1 (PD1) [41], 5T4 oncofetal antigen [42], transmembrane 4 L six family member 1 (TM4SF1) [43], N-glycolyl GM3 ganglioside (NGcGM3) [44], and stage-specific embryonic antigen-4 (SSEA-4) [45].

**Table 1 diagnostics-14-00819-t001:** Target antigens.

	Antigen	Expression in Normal Tissues
Clinical Trials	ALPP	placenta [46]
	B7H3	epithelial cells, pleural effusion, anterior pituitary progenitor cells, and human serum [47]
	c-MET	epithelial cells [48]
	CD70	antigen-activated T and B cells [49]
	Claudin 18	gastric mucosa [50]
	FRα	lung, kidneys, choroid plexus, intestines, heart, and placental tissue [25]
	HER2	nervous system, epithelial cells, or the mammary gland [51]
	HLA-G	embryonic tissues, immune-privileged organs (thymus, cornea, pancreatic islets), hematopoietic lineage (erythroblasts, macrophages, antigen-presenting cells, and dendritic cells) [52]
	Mesothelin	pancreatic, ovarian, mesothelioma cells, and squamous epithelium [20]
	MUC 1	mammary gland, esophagus, stomach, duodenum, pancreas, uterus, prostate lungs, and hematopoietic cells [53]
	MUC 16	Bronchial, endometrial, ovarian, and corneal epithelial cells [30]
	NKG2D	NK cells, CD8+ T cells, γδ T cells, NK1.1+ T cells, and some myeloid cells [54]
	TAG72	secretory endometrial tissue, duodenal goblet cells [55]
Preclinical Studies	5T4 Oncofetal antigen	placenta [42]
	AMHR2	gonads, endometrium, liver parenchyma, epithelial lining of the lung, small intestine mucosa, adrenal, pancreas, and kidney [56]
	ANXA2	endothelial cells, macrophages, and mononuclear cells [57]
	Claudin 6	fetal stomach, fetal kidney, fetal pancreas, and fetal lung [32,33]
	EPCAM	epithelial cells (excluding squamous epithelial, epidermal keratinocytes, gastric parietal cells, myoepithelial cells, thymic cortical epithelium, and hepatocytes), embryonic and hepatic stem cells [58]
	NGcGM3	not present [44]
	PDL-1	antigen-presenting cells, mesenchymal stem cells, and bone marrow-derived mast cells [59]
	PTK7	embryonic tissues, immature CD4+ recent thymic emigrants (RTEs), and plasmacytoid dendritic cells (pDCs) [34,35]
	SSEA-4	embryonic tissues, germ cells [60]
	TM4SF1	endothelium, skin, lung, and germ cells [61]

## 4. Clinical Trials and Outcomes

Currently published clinical trials utilizing CAR-T cells in ovarian cancer patients are summarized below and in Table 2. Most of these trials are phase I, and not designed to evaluate efficacy. Ongoing clinical trials are described in Table 3. In many of these trials, lymphodepletion is used prior to CAR-T cell therapy to alter the tumor microenvironment and attempt to increase CAR-T cell expansion, persistence, and efficacy. Potential mechanisms include decreasing the number of immunosuppressive regulatory T cells (Tregs) and competition from other lymphocytes [62]. The most common regimens in the described trials include cyclophosphamide alone or cyclophosphamide combined with fludarabine.

### 4.1. Mesothelin

In a phase I study, Haas et al. investigated the use of lentiviral-transduced CART-meso cells in fifteen patients with chemotherapy-refractory solid tumors (five patients with malignant pleural mesothelioma, five patients with ovarian carcinoma, and five patients with pancreatic ductal carcinoma). Patients received a single injection of CART-meso cells with or without lymphodepletion with cyclophosphamide. Overall, the treatment was well tolerated, with one dose-limiting toxicity reported. The best result was stable disease (11/15 patients at 28 days, and 3/8 patients at months 2–3) using RECIST 1.1 criteria. One patient with ovarian cancer had a transient appreciable reduction in target tumor burden, but it did not meet RECIST 1.1 criteria for partial response. CART-meso cell expansion and persistence were also evaluated in vivo. In all patients, CART-meso cells expanded and reached peak levels in the peripheral blood by days 6–14. However, persistence was transient, and levels were undetectable in 9 of 15 patients by month 2. Two patients had detectable levels through month six. Lymphodepletion with cyclophosphamide led to a higher expansion of CAR T cells, but it did not increase cell persistence [63].

At the University of Pennsylvania, an ongoing phase I clinical trial is evaluating the use of fully human anti-mesothelin M5 CAR-T cells (NCT03054298). This CAR consists of the M5 single-chain variable fragment (scFV) fused to the costimulatory CD137 (4-1BB) domain) and TCR zeta. Fourteen total patients with mesothelin-expressing tumors, including ovarian, were treated with this M5 CAR with or without lymphodepletion with cyclophosphamide in this basket study. Final data has yet to be published, but preliminary data were presented at the Society of Gynecologic Oncology Annual Meeting in 2022. In the initial cohorts, three patients received a single IV injection of 3 × 10^7^/m^2^ CART-meso cells (cohort 1), three patients received lymphodepletion followed by an IV injection of 3 × 10^7^/m^2^ CART-meso cells (cohort 2), and two patients received an IV injection of 3 × 10^8^/m^2^ CART-meso cells (cohort 3). M5 CART cells expanded in the blood, and levels peaked on days 7–10. The expansion of M5 CART cells was 10-fold higher when lymphodepletion was used. In all patients, M5 CART cells persisted for up to 28–42 days. In two patients, M5 CART cells persisted for six months. Cohorts 1 and 2 were well tolerated without dose-limiting toxicity. However, both patients in cohort 3 experienced serious acute respiratory adverse events. One patient had grade 3 cytokine release syndrome (CRS) and hypoxia, but ultimately recovered. The second patient with mesothelioma and an extensive pulmonary tumor burden died. Upon autopsy, this patient had an accumulation of CART cells in the lungs, extensive T cell infiltration, and acute lung injury [64]. Subsequently, six patients in a new cohort received an initial IV injection of 3 × 10^7^/m^2^ CART-meso cells after lymphodepletion, followed by up to two additional IV doses. In this group, an additional two patients had grade 3 CRS. Stable disease per RECIST 1.1 criteria was seen in 8/14 patients at 28–35 days after treatment. No patients had a clinical response [65].

Chen et al. recently reported three cases of ovarian cancer that were treated with autologous anti-meso CAR-T cells. In two of the three patients, the disease stabilized with progression-free survival times of 5.6 and 4.6 months. There were no reports of CRS or neurologic symptoms over grade 2. CAR-T cells transiently expanded, with reported peaks between 4 and 7 days and persistence of up to 36 days [66]. Fang et al. reported a case of a patient with refractory epithelial ovarian cancer, who had a partial response and 17-month survival after receiving two infusions of autologous CAR-T cells with scFv specific for MSLN and PD-1 (alpha PD-1-mesoCAR-T cells). Infusions were performed on days 0 and 26; the CAR-T copy number in peripheral blood peaked after the second infusion and persisted until the last measurement period on day 53 [67].

In phase I of a clinical trial (NCT03198546), one patient with recurrent stage III ovarian cancer received two intraperitoneal infusions of anti-mesothelin CAR-T cells engineered to secrete IL-7 and CCL19 (anti-MSLN-7x19 CAR-T). She had no serious infusion or therapy-related adverse events, but ultimately had progressive disease on day 38 [68].

Interim data as of February 2023 from an ongoing phase I/Ib clinical trial (NCT03907527) was recently presented at the 2023 American Society of Clinical Oncology meeting. Patients with relapsed or refractory ovarian, fallopian tube, or primary peritoneal cancer were treated with PRGN-3005 UltraCAR-T cells targeting unshed MUC16 and membrane-bound IL-15 via intravenous or intraperitoneal infusion. Twelve patients received intraperitoneal PRGN-3005 UltraCAR-T cells, six patients received IV PRGN-3005 UltraCAR-T cells without lymphodepletion, and seven patients received IV PRGN-3005 UltraCAR-T cells with lymphodepletion (cyclophosphamide). A dose-dependent increase in PRGN-3005 UltraCAR-T cell expansion and persistence in peripheral blood was observed, with and without lymphodepletion. There was observed persistence of PRGN-3005 UltraCAR-T cells of up to 9 months in some patients. In total, 20% of all subjects had a response based on RECIST 1.1 criteria in at least one target lesion. Of patients treated with IV PRGN-3005 UltraCAR-T cells with lymphodepletion, there was a reported 85.7% disease control rate at first restaging, with decreased target tumor burden in 4/7 patients. Overall, the treatment was well tolerated without any dose-limiting toxicities, neurotoxicity, grade 3 or greater CRS, or deaths. The phase Ib expansion stage of the study is ongoing [69].

A phase I/II clinical trial (NCT01583686) using anti-MSLN CAR-T cells in patients with metastatic solid tumors, including ovarian cancer, after pre-treatment with cyclophosphamide and fludarabine was terminated due to insufficient accrual after 6 years. Only one patient out of fifteen had stable disease; the safety profile was manageable [70].

Another phase I clinical trial (NCT03608618) evaluated MCY-M11, a mesothelin-directed CAR mRNA transfected into peripheral blood mononuclear cells, in patients with ovarian carcinoma and malignant peritoneal mesothelioma was terminated due to a sponsor shift in focus [71].

### 4.2. Other

Kershaw et al. published the first phase I trial of CAR-T in ovarian cancer in 2006. Their study demonstrated the safety of a first-generation CAR-T targeting FRα. Of the fourteen patients in the study, eight (cohort 1) received T cell therapy with high dose interleukin-2, and six received dual-specific T cells (reactive to both tumor and allogenic antigen), followed by immunization with allogenic peripheral blood mononuclear cells (cohort 2). CAR-T cells peaked in peripheral blood in the first few days after transfer, and then quickly declined. In all but one patient, levels of CAR-T cells were absent or barely detectable by one month. In cohort 1, five patients experienced grade 3 or 4 toxicities, which were likely attributable to interleukin-2. No grade 3 or 4 toxicities were reported for patients in cohort 2. There was no clinical response to therapy, and the disease progressed for all patients [72].

Preliminary data from a phase I study evaluating the use of TnMUC1 CAR-T cells in patients with advanced TnMUC1-positive solid tumors and multiple myeloma was presented at the 2021 American Society of Clinical Oncology Annual Meeting. At that time, six patients with solid tumors had been treated. There were no reported dose-limiting toxicities. CAR-T cell expansion improved after lymphodepletion chemotherapy. The study has since been terminated by the sponsor due to an unfavorable risk/benefit analysis [73].

**Table 2 diagnostics-14-00819-t002:** Completed clinical trials.

Target	Phase	N	Treatment	Route	Premedication Regimen	Clinical Outcome	Toxicities	Location	Reference
MESO	I	15 (5 Ovarian)	lentiviral transduced CART-meso cells	IV	+/− cyclophosphamide	SD 11/15 at 28 days, 3/8 at 2–3 months	Grade 3: ascites (*n* = 4), fatigue, abdominal pain, abdominal distension, AKI, bacteremia, hepatic failure, hepatitis, dyspnea, DIC, lymphopenia, increased alkaline phosphate, increased ALT, increased AST, increased bilirubin Grade 4: sepsis, anemia	USA	[63]
MESO	–	3 Ovarian	anti-MESO CAR-T cells (LD013)	IV	–	SD 2/3 (PFS 4.6, 5.6 months)	Grade 3/4: none	China	[66]
FRα	I	14 Ovarian	anti-alpha folate receptor CAR-T	IV	–	PD 14/14	Grade 3/4: hypotension (*n* = 4), dyspnea (*n* = 2), fatigue, leukopenia, anemia, thrombocytopenia, rigors, sinus tachycardia, diarrhea	USA	[72]

Abbreviations: MESO (mesothelin), IV (intravenous), SD (stable disease), PD (progressive disease), PFS (progression-free survival), CRS (cytokine release syndrome), AKI (acute kidney injury), DIC (disseminated intravascular coagulation).

**Table 3 diagnostics-14-00819-t003:** Ongoing clinical trials.

Target	Phase	Treatment	Route	Premedication Regimen	Patient Population	N	Status	Study Completion	Location	Clinical Trial Identifier
ALPP	I	retroviral vector-transducted autologous T cells to express anti-ALPP CARs	IV	cyclophosphamide, fludarabine	ALPP-positive metastatic ovarian and endometrial cancer	20	Not Yet Recruiting	12/31/2023 (estimated)	China	NCT04627740
B7H3	I, II	fhB7H3.CAR-Ts	IP	cyclophosphamide, fludarabine	ovarian cancer, especially patients with refractory ascites, with expression of B7H3 antigen in tumor tissue	15	Recruiting	8/31/2026 (estimated)	China	NCT05211557
B7H3	I	CAR.B7-H3 T cells	IP	cyclophosphamide, fludarabine	recurrent epithelial ovarian cancer	21	Recruiting	2/1/2030 (estimated)	USA	NCT04670068
c-Met	I, II	CAR-T cell immunotherapy (anti-c-MET in ovarian cancer patients)	IV	cyclophosphamide, fludarabine	malignancy with positive expression of one of 10 different antigens (c-MET for ovarian cancer patients)	73	Unknown	3/1/2023 (estimated)	China	NCT03638206
CD70	I	CD70 CAR T cells	IV or IP	cyclophosphamide, fludarabine	CD70-positive advanced/metastatic solid tumors	36	Recruiting	5/30/2025 (estimated)	China	NCT05518253
CD70	I	CD70 CAR T cells	IV or IP	cyclophosphamide, fludarabine	CD70-positive advanced/metastatic solid tumors	36	Recruiting	12/31/2024 (estimated)	China	NCT05420545
CD70	I	CD70 CAR T cells	IV or IP	cyclophosphamide, fludarabine	CD70-positive advanced/metastatic solid tumors	48	Recruiting	7/17/2024 (estimated)	China	NCT05468190
CD70	I	CD70 CAR T cells	IV or IP	cyclophosphamide, fludarabine	CD70-positive advanced/metastatic solid tumors	48	Recruiting	9/30/2026 (estimated)	China	NCT06010875
CD70	I, II	anti-hCD70 CAR T cells	IV	cyclophosphamide, fludarabine, aldesleukin	metastatic or unresectable CD70-expressing cancer	124	Recruiting	1/1/2028 (estimated)	USA	NCT02830724
Claudin 18	I	claudin 18.2 CAR-T	IV	yes, not specified	advanced solid tumors with positive CLDN18.2 expression	30	Recruiting	12/1/2024 (estimated)	China	NCT05472857
FRα	I	lentiviral transduced MOv19-BBz CAR T cells	IP	with and without cyclophosphamide, fludarabine	advanced persistent or recurrent high-grade serous epithelial ovarian, fallopian tube, or primary peritoneal cancer-expressing aFR	18	Recruiting	10/2038 (estimated)	USA	NCT03585764
HER2	I	CCT303-406 CAR-modified autologous T cells	IV	cyclophosphamide, fludarabine	relapsed or refractory stage IV HER2-positive cancers	15	Recruiting	3/29/2025 (estimated)	China	NCT04511871
HLA-G	I, II	anti-HLA-G CAR-T cells IVS-3001	IV	cyclophosphamide, fludarabine	previously treated, locally advanced, or metastatic HLA-G-positive solid tumors	117	Recruiting	12/29/2029 (estimated)	USA	NCT05672459
MESO	I	MESO CAR-T cells	IV	cyclophosphamide, fludarabine	relapsed and refractory ovarian cancer	20	Unknown	4/1/2022 (estimated)	China	NCT03799913
MESO	I, II	anti-MESO CAR-T cells	IV	cyclophosphamide, fludarabine	MESO-positive relapsed and refractory epithelial ovarian cancer	20	Unknown	4/20/2023 (estimated)	China	NCT03916679
MESO	I	MSLN-CAR T cells secreting PD-1 nanobodies	IV	cyclophosphamide	advanced solid tumor, positive for MSLN and PDL1 expression	10	Unknown	6/1/2022 (estimated)	China	NCT04503980
MESO	I	LCAR-M23 cells	IV	cyclophosphamide, fludarabine	relapsed and refractory epithelial ovarian cancer	15 (actual)	Terminated	6/7/2022 (actual)	China	NCT04562298
MESO	I	MESO CAR-T cells	IV	cyclophosphamide, fludarabine	mesothelin-positive refractory-relapsed ovarian cancer	10	Unknown	1/1/2023 (estimated)	China	NCT03814447
MESO	I	lentiviral transduced huCART-meso cells	IV or IP	with and without cyclophosphamide	mesothelin-expressing advanced solid tumors	65 (actual)	Active, Not Recruiting	3/1/2025 (estimated)	USA	NCT03054298
MESO	I	anti-mesothelin CAR-T cells	IV	cyclophosphamide, fludarabine, aldesleukin	mesothelin-expressing advanced solid tumors	15 (actual)	Terminated due to slow/insufficient accrual	12/17/2018 (actual)	USA	NCT01583686
MESO	I, II	TCR-like CAR-T	IV	–	mesothelin-expressing advanced ovarian cancer	10	Recruiting	5/16/2025 (estimated)	China	NCT05963100
MESO	I	anti-MESO CAR-T cells	IV	–	relapsed or refractor mesothelin-positive tumors	20	Unknown	11/2018 (estimated)	China	NCT02580747
MESO	I	SynKIR-110	IV	–	recurrent or relapsed advanced ovarian, primary peritoneal, fallopian tube, cholangiocarcinoma, or epithelial mesothelioma-expressing mesothelin	42	Recruiting	3/30/2026 (estimated)	USA	NCT05568680
MESO	I	lentiviral transduced huCART-meso cells in combination with VCN-01	IV	–	persistent or recurrent serous epithelial ovarian carcinoma or unresectable or metastatic pancreatic adenocarcinoma	12	Recruiting	9/2023 (estimated)	USA	NCT05057715
MUC 16	I	PRGN-3005 T cells targetting MUC16	IV or IP	none	recurrent, advanced, platinum-resistant ovarian, fallopian tube, and primary peritoneal cancers	71	Recruiting	11/15/2028 (estimated)	USA	NCT03907527
MUC 16	I, II	A2B694 (an autologous Logic-gated Tmod CART)	IV	yes, not specified	recurrent, unresectable, locally advanced, or metastatic solid tumors that express mesothelin and have lost HLA-A*02 expression	230	Not Yet Recruiting	6/1/2019 (estimated)	USA	NCT06051695
MUC 16	I	4H11-28z/fIL-12/EGFRt+ genetically modified T cells secreting IL-12	IV followed by IP	cyclophosphamide +/− fludarabine	recurrent high-grade serous ovarian, primary peritoneal, or fallopian tube carcinoma-expressing MUC16ecto+	18 (actual)	Active, Not Recruiting	8/2024 (estimated)	USA	NCT02498912
MUC1	I	P-MUC1C-ALLO1	IV	yes, not specified	advanced or metastatic epithelial-derived solid tumors	100	Recruiting	4/2039 (estimated)	USA	NCT05239143
Tn-MUC1	I	CAR-TnMUC1	IV	cyclophosphamide, fludarabine	advanced TnMUC1-positive solid tumors and multiple myeloma	16 (actual)	Terminated due to unfavorable risk/benefit analysis	12/2/2022 (estimated)	USA	NCT04025216
NKG2D	I, II	NKR-2 cells	IV	–	metastatic cancer (multiple)	146	Unknown	8/2021 (estimated)	Belgium, USA	NCT03018405
TAG72	I	TAG72-CAR-T	IP	cyclophosphamide, fludarabine	platinum-resistant epithelial ovarian cancer	33	Recruiting	4/5/2027 (estimated)	USA	NCT05225363

Ongoing clinical trials from ClinicalTrials.gov accessed on 28 December 2023. Abbreviations: ALPP (placental alkaline phosphatase), MESO (mesothelin), IV (intravenous), IP (intraperitoneal).

## 5. Adverse Events

The success of CAR-T cell therapy has been accompanied by several notable adverse events, including CRS, hemophagocytic lymphohistiocytosis/macrophage activation-like syndrome (HLH/MAS), neurotoxicity, and on-target, off-tumor effects. One of the primary concerns in CRS is that the activation and proliferation of CAR-T cells leads to an over-production of pro-inflammatory cytokines, which results in a systemic inflammatory response. It occurs in up to 50–90% of patients and is characterized by symptoms ranging from fever and fatigue to more severe manifestations such as shock, DIC, multi-organ failure, and death [74]. Symptoms typically develop within one week of CAR-T cell infusion, and coincide with the peak of CART cell expansion. CRS is a clinical diagnosis. While it should not guide management, laboratory evaluation demonstrates elevated IL-6, INF-γ, TNF-α, IL-2, ferritin, and CRP [17]. Early recognition and prompt intervention, often with supportive care (acetaminophen, antihistamines, vasopressors, intravenous fluids, and mechanical ventilation), corticosteroids, and anti-cytokine agents like tocilizumab, are important for mitigating the effects of CRS [75].

HLH/MAS is a hyperinflammatory syndrome that is often superimposed with CRS [76]. Clinical features included fever, elevated ferritin, transaminitis, hemophagocytosis, coagulopathy, splenomegaly, and low/absent natural killer cell activity [77]. Management involves supportive care, with consideration of steroids, anakinra (recombinant IL-1 receptor antagonist), or cyclosporine in severe cases [78].

Neurotoxicity is another potentially significant adverse event from CAR-T cell therapy. Immune effector cell-associated neurotoxicity (ICANS) reflects the penetration of CAR-T cells into the central nervous system and manifests as confusion, seizures, and cerebral edema in severe cases. Symptoms typically begin 3–10 days after CAR-T cell administration. Treatment includes supportive care and steroids for severe cases. The underlying mechanism is not fully understood [79].

On-target, off-tumor effects are a significant concern in CAR-T cell therapy, as target antigens are often also expressed on normal tissue [15]. Of antigen targets currently being investigated for ovarian cancer CAR-T cell therapy, several events have been reported in clinical trials for other solid tumors. In patients receiving anti-HER2 CAR T cells, there was a case report of acute respiratory distress syndrome leading to death for a patient with metastatic colon cancer, as well as one case of reversible upper gastrointestinal hemorrhage in a phase I trial of patients with advanced biliary tract and pancreatic cancers [80,81]. Additionally, in a phase I trial of anti-CLDN18.2 CAR T cells in patients with advanced-stage gastric cancer, six of thirty-seven patients developed mucosal toxicity (one with grade 3 toxicity) [82].

In the currently published studies of CAR-T cells in ovarian cancer, no cases of HLH or neurotoxicity have been reported. In an ongoing clinical trial of anti-mesothelin M5 CAR-T Cells (NCT03054298), three patients experienced grade 3 CRS [65]. Two patients experienced serious pulmonary adverse events from on-target off-tumor toxicity, with one patient with mesothelioma and a high pulmonary disease burden dying [64]. Haas et al. reported one patient with dose-limiting toxicity (sepsis) in the cohort treated with 1–3 × 10^7^/m^2^ CAR-T meso without lymphodepletion [63]. In Kershaw et al., only patients who received interleukin-2 with the first-generation CAR-T targeting FRα experienced grade 3 or 4 toxicities [72].

## 6. Future Directions

The clinical trials described above have demonstrated the relative safety and potential therapeutic value of CAR-T cells in ovarian cancer. However, currently, the overall efficacy remains limited. This is likely due, in part, to the immunosuppressive nature of the tumor microenvironment, which contributes to dysfunctional T cell states including exhaustion and senescence [83]. Additional ongoing challenges include limiting on-target, off-tumor effects.

Several strategies are being investigated to overcome these challenges and improve treatment efficacy representing promising future directions. These include loco-regional administration of CAR-T cells, combination therapies, fourth-generation CARs, and KIR CARs.

Loco-regional delivery is being explored to improve safety by limiting on-target and off-tumor effects and improving tracking and tumor infiltration. Data have been published on intrapleural injections in pleural tumors in mouse models and demonstrated greater early accumulation of CAR-T cells at the tumor and eradication of pleural tumors at lower CAR-T doses [84]. Several ongoing clinical trials evaluating intraperitoneal infusions of CAR-T cells in ovarian cancer are described in Table 2.

Combination therapies combine CAR-T cell therapy with chemotherapy, oncoviruses, checkpoint inhibitors, immunomodulatory agents, or other therapies to enhance safety and efficacy [85]. Currently, there is an ongoing phase I clinical trial evaluating the oncovirus VCN-1 administered in combination with human CAR-T cells targeting mesothelin in patients with unresectable or metastatic pancreatic or persistent or recurrent ovarian cancer (NCT05057715). VCN-1 is an adenovirus designed to replicate in cells with a dysfunctional RB1 pathway. It expresses a soluble human recombinant hyaluronidase to enhance the intratumor spread of the virus by degrading the extracellular matrix and disrupting tumor stroma, facilitating immune cell penetration into the tumor [86]. Another phase I clinical trial is registered in China utilizing IV MSLN-CAR T cells secreting PD-1 nanobodies after pre-treatment with cyclophosphamide for patients with advanced solid tumors expressing mesothelin and PDL1 (NCT04503980).

Fourth-generation CARs, also called TRUCK T cells (“T cells redirected for antigen-unrestricted cytokine-initiated killing”), are engineered to produce cytokines or antibodies [13]. They allow for direct depositing of pro-inflammatory cytokines to the target tissue while minimizing systemic inflammatory effects. Investigated cytokines include IL-12, which activates T cells and NK cells, induces the production of interferon-gamma, and assists in the differentiation of helper T cells [87]. A clinical trial is currently ongoing in the United States (NCT02498912) utilizing genetically modified T cells secreting IL-2 and targeting the MUC16ecto antigen in patients with recurrent high-grade serous ovarian, primary peritoneal, or fallopian tube carcinoma-expressing MUC16ecto. Their study incorporates both intravenous and intraperitoneal administration in a standard dose escalation, with and without pre-treatment with cyclophosphamide and fludarabine. In preliminary data presented at the Society of Gynecologic Oncology 2020 Annual Meeting, eighteen patients were enrolled. The best response was stable disease. Two out of three patients in the lymphodepletion cohort experienced dose-limiting toxicities (HLH/MAS). There were no reported dose-limiting toxicities in the other cohorts, but CRS was seen at all doses [88].

Another novel CAR-T cell therapy is the KIR-CAR platform being developed by Verismo Therapeutics. The KIR-CAR platform includes dual-chain CAR-T cell therapy, including independent binding and signal chains that couple when the tumor is engaged and a DAP12 signal for activation and co-stimulation of T cells. In preclinical studies, this has improved KIR-CAR T cells’ efficacy period [89]. A phase I clinical trial using intravenous SynKIR-110 is registered in the United States (NCT05568680) for patients with mesothelin-expressing ovarian cancer, mesothelioma, and cholangiocarcinoma, though the status is currently unknown.

## 7. Conclusions

CAR-T cell therapy involves the use of genetically modified T cells expressing a synthetic receptor to target cancer-specific antigens. While still in the initial stages of development, CAR-T cell therapy has led to remarkable results in hematologic malignancies and is a promising potential therapeutic in ovarian cancer. Ongoing challenges include the short persistence of CAR-T cells, inter- and intratumor heterogeneity, antigen escape, and the immunosuppressive tumor. Numerous target antigens including mesothelin, MUC16, FRα, and HER2 have been studied in the preclinical setting, and now are being brought to clinical trials. Early phase I data has demonstrated the safety of CAR-T cell therapy using constructs targeting mesothelin and FRα, however lasting clinical responses have been limited. As use expands, the recognition and timely management of adverse events including CRS, neurotoxicity, and HLS/MAS is essential to ensuring patient safety. Several strategies to increase the therapeutic efficacy while minimizing adverse events are being explored, including loco-regional delivery of CAR-T cells, combination therapies, fourth-generation CARs, and KIR-CARs.

Combination strategies with oncovirus, or targeted agents that can modify tumor microenvironment and create a more appropriate immune milieu that permits CAR-T infiltration, expansion, and functionality may prove successful in the near future. In addition, next-generation ON-OFF switch CAR-T designs can mitigate side effects seen with traditional designs by limiting inflammatory cytokine production while maintaining anti-tumor efficacy.

## Figures and Tables

**Figure 1 diagnostics-14-00819-f001:**
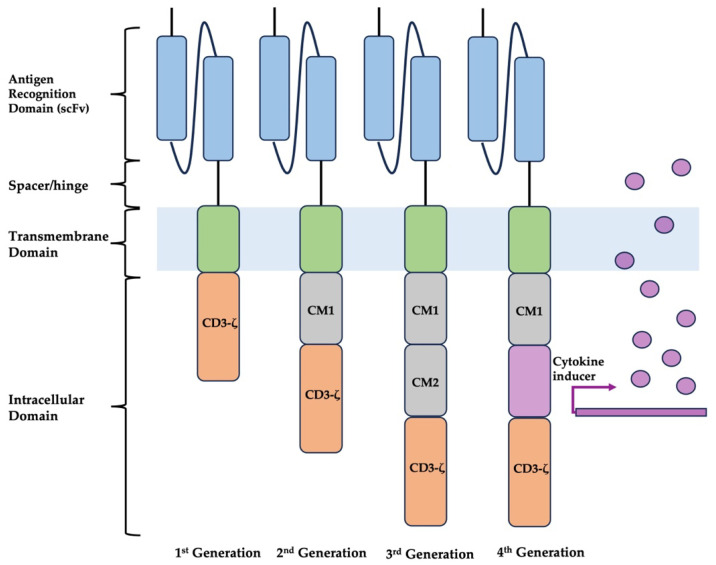
CAR generations. Abbreviations: ScFv: single-chain variable fragment, CM: costimulatory domain.

## Data Availability

There are no additional data deposited on any other site other than in this manuscript.

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
