# Peer review of "CAR-T Cell Therapy in Ovarian Cancer: Where Are We Now?"

_diagnostics, 2024, doi:10.3390/diagnostics14080819_

Round 1

Reviewer 1 Report

Comments and Suggestions for Authors

The authors propose a review on CAR-Ts in the treatment of ovarian cancer.

-The introduction could be expanded with the state of the art at least on immunotherapy for ovarian cancer and on the current role of adoptive cell therapies in this disease.

Bibliographic references are missing in lines 27 and 29.

-          Car-T cell design is very scholastic and could be summarized by a figure that would be more impactful

-          A summary table of the results of completed studies including toxicities would be helpful.

The section on adverse events is again very general, scholastic, and focused on treatments in oncohematology. The topic of on-target off tumor toxicities, which is very much present in solid tumors, is not touched on at all. Also, an examination of whether there are differences in toxicities for example on the type of lymphodepletion or post infusion adjuvant of CAR-T could be useful. A comparative table might be also useful.

It would be appreciable to understand the authors' views on which future strategies are truly promising or which CAR-T constructs might actually be successful.

The paragraph is a bit of a repeat of the list of ongoing clinical trials and not an examination of the strategies that could be put in place as researchers in the field.

Author Response

Comment 1: The introduction could be expanded with the state of the art at least on immunotherapy for ovarian cancer and on the current role of adoptive cell therapies in this disease.

Reply to comment 1: Introduction has been expanded and summarized the use of immunotherapies in ovarian cancer- including checkpoint inhibitors, cancer vaccines in ovarian cancer, and adoptive cell transfer.

Comment 2: Bibliographic references are missing in lines 27 and 29.

Reply to comment 2: References have been added.

Comment 3: Car-T cell design is very scholastic and could be summarized by a figure that would be more impactful.

Reply to comment 3: Figure has been added to summarize the different generations of Car-T cells.

Comment 4: A summary table of the results of completed studies including toxicities would be helpful.

Reply to comment 4: Table 2 has been expanded to include results and grade 3/4 toxicities for completed studies.

Comment 5: The section on adverse events is again very general, scholastic, and focused on treatments in oncohematology. The topic of on-target off tumor toxicities, which is very much present in solid tumors, is not touched on at all. Also, an examination of whether there are differences in toxicities for example on the type of lymphodepletion or post infusion adjuvant of CAR-T could be useful. A comparative table might be also useful.

Reply to comment 5: An additional paragraph has been added to discuss on-target off-tumor toxicities reported in other solid tumors (focus on published results using same targets being investigated for ovarian cancer). Discussion of adverse events expanded to highlight significant events and associated treatment regimens (including adjuncts, and lymphodepletion) in published literature for CAR-T cells and ovarian cancer.

Comment 6: It would be appreciable to understand the authors' views on which future strategies are truly promising or which CAR-T constructs might actually be successful. The paragraph is a bit of a repeat of the list of ongoing clinical trials and not an examination of the strategies that could be put in place as researchers in the field.

Reply to comment 6: we have modified the conclusion and provide our view on which strategies are promising. (lines 349-356)

Reviewer 2 Report

Comments and Suggestions for Authors

In the present manuscript authors explore the feasibility of adapting CAR-T therapies for ovarian cancer treatment, drawing insights from clinical studies and publications in the field. They also address significant hurdles faced by CAR-T therapies, such as the immunosuppressive tumor microenvironment and therapy-related toxicities.

Manuscript is well written, but may benefit from these minor revisions:

1) Ensure consistency in headlines 3.1-3.4 by using either gene names, or gene symbols;

2) Reformat Table 2 to include key toxicity and efficacy indicators, along with the number of ovarian cancer patient in the study;

3) A potentially confusing sentence on line 181, clarify that it was anti-MSLN CAR-T cells;

4) Consider extending the discussion on lines 261-262, referencing an additional resource for further exploration (10.3390/cancers14041078).

Author Response

Comment 1: Ensure consistency in headlines 3.1-3.4 by using either gene names, or gene symbols.

Reply to comment 1: Headlines updated to all gene names.

Comment 2: Reformat Table 2 to include key toxicity and efficacy indicators, along with the number of ovarian cancer patient in the study.

Reply to comment 2: per reviewer’s suggestion Table 2 has been expanded to include results, grade 3 and 4 toxicities, and number of ovarian cancer patients in the studies.

Comment 3: A potentially confusing sentence on line 181, clarify that it was anti-MSLN CAR-T cells.

Reply to comment 3: Sentence has been edited to specify anti-MSLN CAR-T cells.

Comment 4: Consider extending the discussion on lines 261-262, referencing an additional resource for further exploration (10.3390/cancers14041078).

Reply to comment 4: Discussion has been expanded and now includes suggested reference.